# Heat shock proteins (HSP 90, 70, 60, and 27) in *Galleria mellonella* (Lepidoptera) hemolymph are affected by infection with *Conidiobolus coronatus* (Entomophthorales)

**Anna Katarzyna Wrońska** [1] *, **Mieczysława Irena Boguś** [1,2]

**1** Polish Academy of Sciences, Witold Stefański Institute of Parasitology, Warsaw, Poland, **2** BIOMIBO, Warsaw, Poland

* a.wronska@twarda.pan.pl

**Data Availability Statement:** All relevant data are within the manuscript and its Supporting Information files.

## Abstract

Invertebrates are becoming more popular models for research on the immune system. The innate immunity possessed by insects shows both structural and functional similarity to the resistance displayed by mammals, and many processes occurring in insect hemocytes are similar to those that occur in mammals. The humoral immune response in insects acts by melanization, clotting and the production of reactive oxygen species and antimicrobial peptides, while the cellular immunity system is based on nodulation, encapsulation and phagocytosis. An increasingly popular insect model in biological research is *Galleria mellonella*, whose larvae are sensitive to infection by the entomopathogenic fungus *Conidiobolus coronatus*, which can also be dangerous to humans. One group of factors that modulate the response of the immune system during infection in mammals are heat shock proteins (HSPs). The aim of this study was to investigate whether infection by *C. coronatus* in *G. mellonella* hemolymph is accompanied by an increase of HSP90, HSP70, HSP60 and HSP27. Larvae (five-day-old last instar) were exposed for 24 hours to fully-grown and sporulating fungus. Hemolymph was collected either immediately after termination of exposure (F24) or 24 hours later (F48). The concentration of the HSPs in hemolymph was determined using ELISA. Immunolocalization in hemocytes was performed using fluorescence microscopy and flow cytometry. HSP90, HSP70, HSP60 and HSP27 were found to be present in the *G. mellonella* hemocytes. HSP60 and HSP90 predominated in healthy insects, with HSP70 and HSP27 being found in trace amounts; HSP60 and HSP27 were elevated in F24 and F48, and HSP90 was elevated in F48. The fungal infection had no effect on HSP70 levels. These findings shed light on the mechanisms underlying the interaction between the innate insect immune response and entomopathogen infection. The results of this innovative study may have a considerable impact on research concerning innate immunology and insect physiology.

**Funding:** This work was partly supported by two grants: National Centre for Research and Development grant POIG.01.04.00-14-019/12 and Masovian district grant RPMA.01.02.00- 14-5626/16-00 to the Biomibo company. There was no additional external funding received for this study. The funders provided support in the form of salaries for author MIB, but did not have any additional role in the study design, data collection and analysis, decision to publish, or preparation of the manuscript. The specific roles of these authors are articulated in the 'author contributions' section.

**Competing interests:** The authors have read the journal's policy and have the following conflicts: MIB is the President of Biombio. This does not alter our adherence to all the PLOS ONE policies on sharing data and materials. There are no patents, products in development or market products to declare.

## Introduction

Over the past decades, immunological studies have tended to favor the use of murine and rat models. However, as the use of such animals is expensive and laborious, and large numbers of animals have to be maintained to obtain statistically relevant data, there is a growing need for other models such as those based on invertebrates [1–4]. Comparative genome studies have identified numerous homologues to human genes coding for proteins involved in pathogen recognition or signal transduction in insects and other invertebrates. Hence, studies on the virulence of microorganisms and host immunity are increasingly using models based on insects such as *Drosophila melanogaster*, *Blattella germanica*, *Galleria mellonella*, *Culex quinquefasciatus* and *Bombyx mori* [5].

An increasingly popular model in biological research is *G. mellonella*, also known colloquially as the wax worm/moth. The larvae have considerable advantages as models: they are easily and inexpensively obtained in large numbers, and are simple to use and easy to maintain without special laboratory equipment [6]. The fact that the final instar larvae measures 12-20mm is also important, as this allows easy manipulation and facilitates the collection of tissue/hemolymph samples for analysis. In addition, it is easy to administer test substances to the larvae with food, topical preparations or through injection. The short life cycle of *G. melonella*, i.e. seven to eight weeks, also makes them perfect for large-scale studies, with the length of the cycle also being regulated by changing the temperature; in addition, the female wax moth is able to deposit 1500 eggs. The temperature at which these insects can be grown is very important. In contrast to many other alternative invertebrate models, they can grow in a wide temperature range (18–37˚C) [7–10]. This feature is useful in immunological research because it allows to conduct tests at a temperature of 37˚C prevailing in the body of mammals.

*G. mellonella* is also a model host for human pathogens like *Bacillus cereus* [11], *Candida albicans* [12], *Listeria monocytogenes* [13] or *Staphylococcus aureus* [14]. This insect is also used in studies of entomopathogenic fungi [15–17]. *Conidiobolus coronatus* (Entomophthorales) is a soil fungus that is pathogenic to insects [18] and sometimes also to humans. It is known to cause chronic infection in immunocompetent patients, especially in a hot climate. *C. coronatus* is also known as rhinofacial mycosis due to its potential to invade the adjacent skin and the subcutaneous tissue of the face and nose, resulting in deformity [19, 20]. To prevent infection and learn how to manage its effects, it is therefore important to understand the action of the immune system during infection.

The immunological system of *G. mellonella* larvae shows great structural and functional similarity to the innate immune response of mammals: the insect cuticle acts as barrier to pathogens in a similar way as the mammalian skin and insect hemolymph can be partly compared to blood insofar that both tissues contain immunocompetent cells [21]. Although insects have not developed the acquired immunity of mammals, which requires the production of specific antibodies, they are still able to synthesize a series of analogous antimicrobial peptides (AMPs) and secrete them to the hemolymph [22]. The humoral immune response can also be realized through melanization, clotting and the production of reactive oxygen species (ROS).

The cellular immunity system in *G. mellonella* is based on phagocytosis, nodulation and encapsulation reactions and is associated with the occurrence of five types of hemocytes with different functions in the immune response. Plasmatocytes and granulocytes, the most predominant cells, are reported to be active phagocytes; in contrast, oenocytoids, spherulocytes and prohemocytes are less studied and appear to play only minor roles in the immune response [23–26].

The factors regulating the immune response in mammals are relatively well understood. In insects, these issues are poorly described in the literature. The humoral immune responses of

insects mainly involves the release of AMPs by the fat body, via the Toll, the Imd (immune deficiency), and the JAK-STAT (Janus kinase-signal transducer and activator of transcription) pathways. Gram-positive bacteria and fungi predominantly induce the Toll signaling pathway, whereas Gram-negative bacteria activate the Imd pathway. The action of the immune system of insects is also regulated by hormones and neuropeptides [27–29]. For example, it is thought that serotonin may regulate phagocytosis [30].

As the same regulatory pathways are described in mammals, it is worth investigating whether other factors regulating the immune system of mammals may also be present in insects. One such group of factors comprises the heat shock proteins (HSPs). It has been suggested that HSPs represent the link between the innate and adaptive immune systems. HSPs have been reported to play important roles in antigen presentation, lymphocyte and macrophage activation, and dendritic cell activation and maturation [31]. HSP90, HSP70 and HSP60 have the greatest impact on the functioning of the mammalian immune system while HSP27 has slightly less [32–34].

Studies have confirmed the present of heat shock proteins in *G. mellonella* [35, 36]. Therefore, the aim of this study was to investigate whether infection with *C. coronatus* is associated with an increase in the levels of HSP90, HSP70, HSP60 and HSP27 in *G. mellonella* hemolymph.

## Materials and methods

### Insects

A culture of the wax moth, *Galleria mellonella* (Lepidoptera: Pyralidae) was maintained and reared in temperature and humidity-controlled chambers (30˚C, 70% r.h.) in constant darkness on an artificial diet [37]. Fully-grown larvae were collected before pupation, surface-sterilized and homogenized, and then used as a supplement in the fungal cultures. Five-day-old last instar larvae were used for analyzing the influence of fungal infection on HSP in insect hemocytes.

### Fungus

*Conidiobulus coronatus* (isolate number 3491), originally isolated from *Dendrolaelaps* spp., was obtained from the collection of Prof. Bałazy (Polish Academy of Sciences, Research Center for Agricultural and Forest Environment, Poznań). It was routinely maintained in 90 mm Petri dishes at 20˚C with cyclic changes of light (L:D 12:12) on Sabouraud agar medium (SAM) with the addition of homogenized *G. mellonella* larvae to a final concentration of 10% wet weight. The sporulation and virulence of the SAM *C. coronatus* cultures was enhanced with the addition of homogenized *G. mellonella* larvae.

### Infection of insects with *C. coronatus*

*G. mellonella* larvae (five-day-old last instar) were exposed for 24 hours at a temperature of 20˚C to fully-grown and sporulating *C. coronatus* colonies. Fifteen individuals were maintained in each Petri dish. A control group was formed of larvae exposed for 24 hours to sterile Sabouraud agar medium (Merck). After exposure, the insects were transferred to new, clean Petri dishes with appropriate food (an artificial diet [37]) and kept at 20˚C for one day. Following this 24-hour exposure to the fungus, one group of insects was collected immediately for examination (F24 group) while the rest were left for another 24 hours before collection (F48 group).

## Larval hemolymph collection

*G. mellonella* hemolymph was collected from both control and infected (F24 and F48) larvae. Due to the high mortality of insects (65±5,6% in F24 and 87±4,8% in F48), hemolymph was collected from both living and dead individuals. Before bleeding, larvae were washed with distilled water and then immersed briefly in 70% (v/v) ethanol to sterilize their surfaces, thus reducing the contamination of hemolymph samples. Hemolymph was taken from the larvae through an incision made in the last proleg. The hemolymph was prepared in different ways depending on the planned method.

For hemocyte culture, 100 μl of fresh hemolymph collected from ten larvae was suspended in 500 μl of supplemented Grace's Insect Medium (GIM; Invitrogen) with added gentamycin (10mg/ml; Gibco), amphotericin B (250μg/ml; Gibco) and phenylothiourea (PTU; 0.1mM; Sigma-Aldrich). Then, it was transferred to a six-channel μ-Slide IV 0.4 (IBIDI)- 100μl for each channel. The slides were incubated in 30˚C for 24 hours.

For ELISA (enzyme-linked immunosorbent assay) 100 μl of fresh hemolymph collected from ten larvae were suspended in 100 μl of supplemented GIM. Samples were sonicated (20 kHz, 3 min) for cell lysis. Probes were centrifuged at 10,000 x g for 10 min to pellet debris. Supernatants were transferred to a new microcentrifuge tube and stored at -20˚C.

For flow cytometric analysis, 100 μl of fresh hemolymph collected from ten larvae was suspended in 100 μl of supplemented GIM with 10mM EDTA (ethylenediaminetetraacetic acid) and 30mM sodium citrate.

## Immunocytochemical analysis

Immunolocalization of HSP 90, 70, 60 and 27 was performed in all hemocyte cultures (controls, F24 and F48). The cells were fixed in 4% Paraformaldehyde (Sigma-Aldrich; PFA) in Phosphate-buffered saline (PBS) and permeabilized in 0.1% Triton X-100 (Sigma-Aldrich) in PBS. The cells were incubated overnight at 4˚C with primary antibodies all from Enzo Life Sciences: HSP90α/β monoclonal antibody, HSP70 monoclonal antibody, HSP60 (insect) polyclonal antibody or HSP27 polyclonal antibody. Antibodies were diluted 1:40 in PBS with 1% bovine serum albumin (BSA, Sigma-Aldrich). The cells were then incubated in 4% BSA-PBS for two hours to prevent non-specific antibody binding. Following this, the hemocytes were incubated for a further two hours at room temperature with secondary antibody DyLight 488, Goat Anti-Mouse IgG (Abbkine) for HSP 90 and 70 or Dylight 594, Goat Anti-Rabbit IgG (Abbkine) for HSP 60 and 27. Concentrations of secondary antibodies were 2μg/ml. Actin-Green 488 ReadyProbes Reagent (Invitrogen) or ActinRed 555 ReadyProbes Reagent (Invitrogen) were used to label the actin fibers. The cell nuclei were stained with Hoechst (Enzo Life Sciences). Fluorescence signals were analyzed by fluorescent microscopy an Axio Vert.A1 fluorescence microscope (Zeiss) with Axio Cam ICc 5 (Zeiss).

## Heat shock proteins quantification by ELISA

Quantitative HSP analysis was carried out using ELISA tests all from Enzo Life Sciences. The following commercial kits were used: HSP27 (human) ELISA kit, HSP60 (human) ELISA kit, HSP70 high sensitivity ELISA kit and HSP90 (human) ELISA kit. Each test was performed in three independent replicates according to the manufacturer's instructions.

## Flow cytometry analysis

The collected hemolymph (from control and infected larvae) was centrifuged (400xg, 10 min) and washed with PBS. The cells were fixed in 4% Paraformaldehyde (Sigma-Aldrich; PFA) in

phosphate-buffered saline (PBS) and permeabilized in 0.1% Triton X-100 (Sigma-Aldrich) in PBS. The cells were incubated with primary antibodies (diluted 1:100) overnight at 4˚C. The same antibodies were used as described above for HSP immunolocalization. After three washes in PBS, the cells were incubated (two hours at room temperature) with the secondary antibodies: DyLight 488, Goat Anti-Mouse IgG (Abbkine) for HSP 90 and 70 or Dylight 488, Goat Anti-Rabbit IgG (Abbkine) for HSP 60 and 27. Readings were acquired on an CyFlow Cube 8 (Sysmex) and analyzed with FCS Express 6 (DeNovo Software). For each experimental condition, 100µl of each sample was scrutinized. Data was acquired using a 488 nm laser for the detection of each HSP on the FL-1 channel. Results were shown in two forms: histograms displaying a single measurement parameter (relative FITC fluorescence) on the x-axis and the number of events (cell count) on the y-axis, and as dot plots comparing forward scatter (FSC) with side scatter (SSC).

## Statistics

The normality of the data was tested using the Kolmogorov-Smirnov test. The t-test for independent samples was used to compare the results of the control group and the study group. The results were regarded as being statistically significant at $p \leq 0.05$. STATISTICA 6.1 software (StatSoft Polska) was used for all calculations.

## Results

The effects of *C. coronatus* infection on the amounts of four HSPs (90, 70, 60, and 27) in hemocytes were examined in three groups of *G. mellonella* larvae. Larvae were exposed to the fungus for 24 hours: of these, one group was collected immediately afterwards (F24 group) and the rest collected 24 hours later, i.e. 48 hours after initial exposure (F48 group). In addition, another group was exposed for 24 hours to sterile Sabouraud agar medium (controls). Following collection, all samples were subjected to examination by fluorescence microscopy, flow cytometry and ELISA.

Fluorescent images show stained β- actin fibers, cell nuclei, immunolocalized HSPs and merged photos. Two types of hemocytes were visible in the microscopic images: plasmatocytes and granulocytes. The remaining hemocyte subpopulations of *G. mellonella* i.e. spherulocytes, oenocytois and prohemocytes, are not adherent and were washed out during the fixation and staining procedures.

Flow cytometry data was presented as histograms of FITC intensity versus cell count, and as dot plots of FSC (forward scatter) versus SSC (side scatter). All cells present in the sample are marked as gate 1 (red), while cells that contain the tested protein were marked as gate 2 (green). Although all cell subpopulations were examined by this method, it was not possible to determine individual hemocyte subpopulations. ELISA tests were performed using full hemolymph (lysed cells and plasma). Results are shown in the graphs as mean with standard deviation (SD). Raw data from ELISA tests were presented in S2 Table.

### Heat shock protein 90

HSP 90 was detected in *G. mellonella* hemocytes (Fig 1). A slight fluorescence was found on the Texas Red channel in the microscopic images in the control and F24 groups. A much stronger fluorescent signal was observed in the F48 group (Fig 1A). Quantitative ELISA tests showed 0.54 ng/ml of HSP 90 in healthy controls, 0.46 ng/ml in F24 and 2.11 ng/ml in F48: significantly more was observed in F48 than controls (p<0.001) (Fig 1B).

The F48 group displayed the most intense green fluorescence (gate 2) according to flow cytometry analysis (Fig 1C). Analysis of the dot plots found the percentage of cells containing

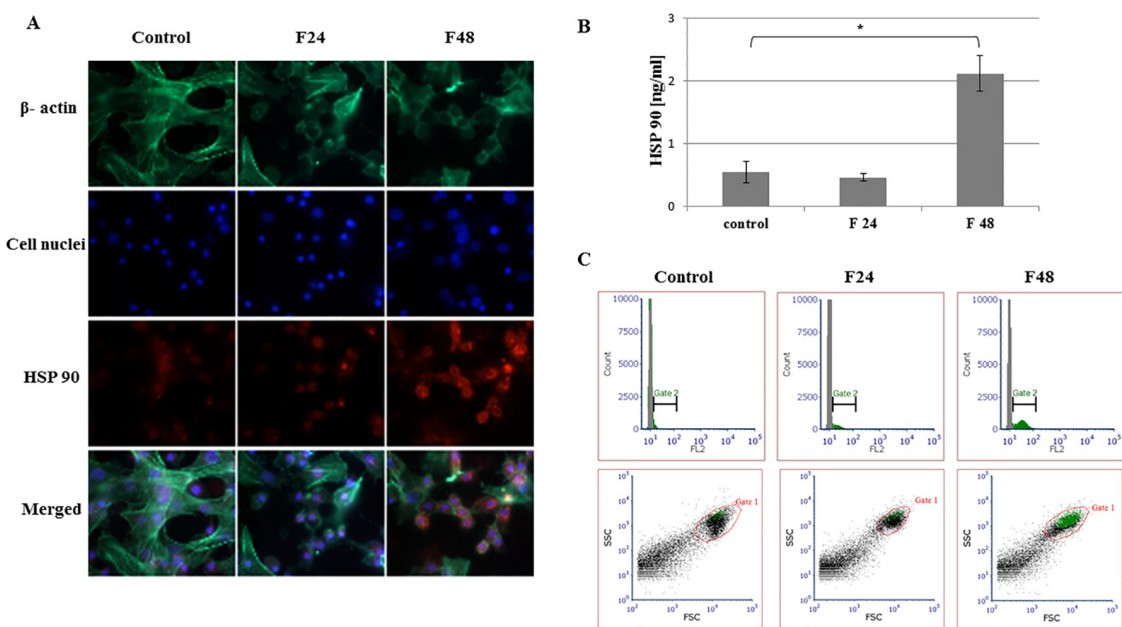

**Fig 1. HSP 90 in hemocytes from *G. mellonella* larvae infected with *C. coronatus*.** (A) Immunofluorescence localization of HSP 90 in hemocytes. β- Actin (green) was stained by ActinGreen 488 ReadyProbes Reagent (Invitrogen). Cell nuclei (blue) were stained with Hoechst (Enzo Life Sciences). HSP 90 (red) was detected using HSP 90α/β monoclonal antibody (Enzo Life Sciences) and Dylight 594, Goat Anti-Rabbit IgG (Abbkine). (B) HSP 90 concentration in hemolymph as determined by ELISA (Enzo Life Sciences). (C) Flow cytometry data as histograms (FITC intensity versus cell count) and dot plots (FSC versus SSC). Gate 1 (red)- all cells, gate 2 (green)- cells containing HSP 90. F24- larvae sampled immediately after 24-hour exposure to fungal infection; F48- larvae sampled 24 hours after 24-hour exposure;*p≤0.05; scale bar 25µm.

HSP 90 to be 13.2±2.9% in controls and 14.5±3.8% in F24. Significantly more hemocytes containing HSP90 (59.6±5.7%) were found in F48 insects (p<0.001) (S1 Table).

### Heat shock protein 70

The effect of *C. coronatus* fungal infection on HSP 70 level in *G. mellonella* hemocytes is presented in Fig 2. The low fluorescence intensity on the Texas Red channel confirms the presence of a small amount of this protein in both the control and test samples (Fig 2A). Quantification tests (ELISA) showed HSP 70 to be present in controls (0.045±0.006 ng/ml), F24 (0.054±0.007 ng/ml) and F48 (0.055±0.007 ng/ml). No statistically significant differences were found between these groups (Fig 2B). These findings were confirmed by flow cytometry (Fig 2C), which indicated similar percentages of HSP 70-positive cells in all tested groups: 11.49 ± 0.59% (control), 12.88 ± 1.16% (F24), 11.29 ± 0.93% (F48) (S1 Table).

### Heat shock protein 60

HSP 60 level increased in *G. mellonella* hemocytes following *C. coronatus* infection. The photos obtained by fluorescence microscopy revealed a clear increase in green fluorescence intensity in groups F24 and F48. This protein is also present in the hemocytes of healthy larvae, as confirmed by fluorescence on the FITC channel in the control group (Fig 3A). Although the highest level of HSP 60 was recorded in F48 (1.61 ng/ml), the level in F24 was also significantly higher than controls (1.11 ng/ml) (Fig 3B). These findings were confirmed by flow cytometry (Fig 3C): the histograms show an increase in fluorescence intensity (gate 2) in groups F24 and

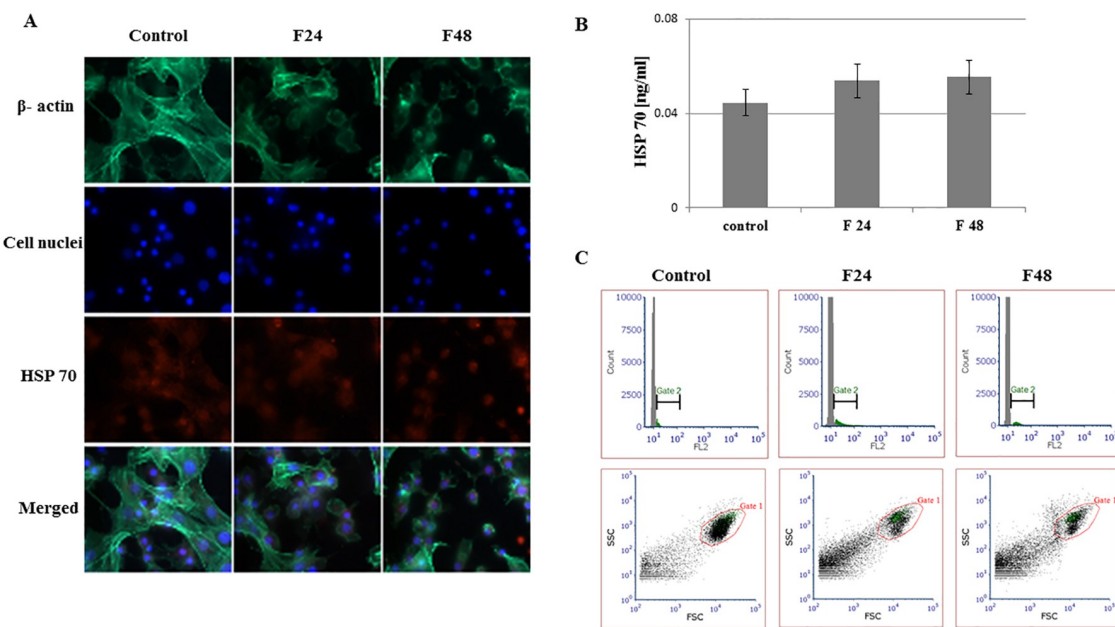

**Fig 2. HSP 70 in hemocytes from *G. mellonella* larvae infected with *C. coronatus*.** (A) Immunofluorescence localization of HSP 70 in hemocytes. β- Actin (green) was stained by ActinGreen 488 ReadyProbes Reagent (Invitrogen). Cell nuclei (blue) were stained with Hoechst (Enzo Life Sciences). HSP70 (red) was detected using HSP70 monoclonal antibody (Enzo Life Sciences) and Dylight 594, Goat Anti-Rabbit IgG (Abbkine). (B) HSP70 concentration in hemolymph, determined by ELISA (Enzo Life Sciences). (C) Flow cytometry data presented as histograms (FITC intensity versus cell count) and dot plots (FSC versus SSC). Gate 1 (red)- all cells, gate 2 (green)- cells that contain HSP70. F24- larvae sampled immediately after 24-hour exposure to fungal infection; F48- larvae sampled 24 hours after 24-hour exposure; scale bar 25μm.

F48 and the dot plot reveals the subpopulations of cells in which HSP 60 is present. The percentage of cells containing the test protein is shown in the S1 Table.

## Heat shock protein 27

HSP 27 was observed in both tested groups of hemocytes. Microscopic observations showed fluorescence on the FITC channel in F24 and F48, and very weak green fluorescence in the control group (Fig 4A). The highest concentration was detected in F24 (0.11ng/ml). A smaller, yet still significant, increase was noted in the F48 group (0.09ng/ml) (Fig 4B). These results are confirmed by the flow cytometry data. Gate statistical analysis found the percentages of cells containing HSP 27 to be 15.3 ± 1.8% in F24 and 19.4 ± 1.2% in F48; both values were higher than in controls (10.4±1.1%) (S1 Table).

It can be concluded that in hemocytes of *G. mellonella* larvae HSP90, HSP70, HSP60 and HSP27 are present. In healthy insects (control), the highest concentration was recorded for HSP60 and HSP90, whereas HSP70 and HSP27 were found in trace amounts. HSP60 and HSP27 level were elevated in the F24 and F48 larvae, and HSP90 was also elevated in the F48 group. The fungal infection had no effect on HSP70 levels.

## Discussion

Insects are often used in biological research and can replace mammals in studies on the immune system or host-pathogen interactions. One such insect is the wax moth, *G. mellonella*. These insects are easy to grow: they have a short life cycle, they easily reproduce, and do not have high nutritional requirements. Wax moths can be cultured in a wide range of

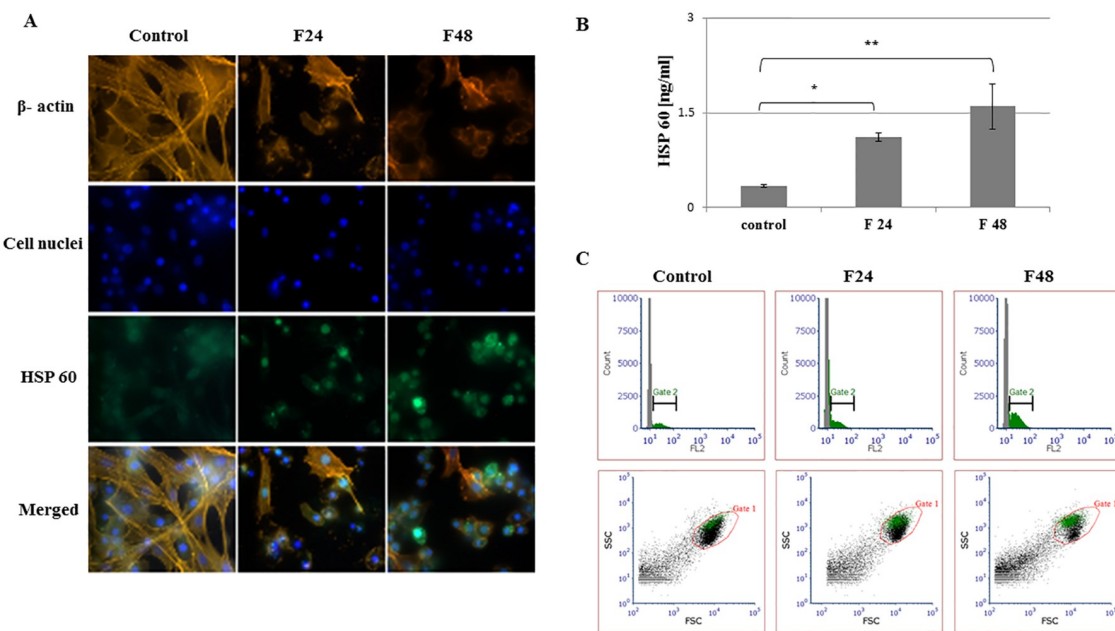

**Fig 3. HSP 60 in hemocytes from *G. mellonella* larvae infected with *C. coronatus*.** (A) Immunofluorescence localization of HSP60 in hemocytes. β- Actin (orange) was stained by ActinRed 555 ReadyProbes Reagent (Invitrogen). Cell nuclei (blue) were stained with Hoechst (Enzo Life Sciences). HSP60 (green) was detected using HSP60 monoclonal antibody (Enzo Life Sciences) and Dylight 488, Goat Anti-Rabbit IgG (Abbkine). (B) HSP60 concentration in hemolymph, as determined by ELISA (Enzo Life Sciences). (C) Flow cytometry data presented in histograms (FITC intensity versus cell count) and dot plots (FSC versus SSC). Gate 1 (red)- all cells, gate 2 (green)- cells that contain HSP60. F24- larvae sampled immediately after 24-hour exposure to fungal infection; F48- larvae sampled 24 hours after 24-hour exposure;$^*$p≤0.05; $^{**}$p≤0.01; scale bar 25μm.

temperatures (18–37˚C), which allows for mapping different research systems. The present study uses *G. mellonella* as a model for the study of fungal infections. Larvae (five-day-old last instar) were infected with *C. coronatus* for 24 hours.

Our previous studies showed that *G. mellonella* larvae are susceptible for *C. coronatus* infection, with the majority of treated larvae dying 24 or 48 hours following contact with the fungal pathogen (79 and 92%, respectively). In the control group comprising larvae which had no contact with the fungus, mortality was much lower (16%) [38].

*C. coronatus* is a saprotrophic fungus which is known mainly as an entomopathogen. Boguś et al. [18] found it to selectively infect different species of insects, and to show strong virulence against *G. mellonella*. It enters the host body cavities through the cuticle by a combination of mechanical pressure from the growing hyphae and enzymatic degradation of major cuticle components, i.e. proteins, chitin and lipids. Inside the insect, the fungus propagates, consuming nutrients and releasing metabolites, resulting in mycosis and, ultimately, host death. Insects have developed a number of defense strategies against fungi; however, the main ones are the presence of a cuticle and a well-developed immune system.

*C. coronatus* is also able to infect higher animals and humans: the first human infection was reported in Jamaica in 1965 [39]. It is a causative fungal agent of chronic rhino facial zygomycosis and chronic, long-standing infection can lead to morbidity [20]. Therefore, it is important to understand the processes by which *C. coronatus* gains entry to the host, as not only can the fungus and its metabolites be used to control insect pest populations, a more accurate understanding of the mechanisms of infection can improve treatment in humans.

Like bacterial or viral infection, fungal infection is stressful for the organism. This stress activates a series of defense reactions, one of which involves the production of heat shock

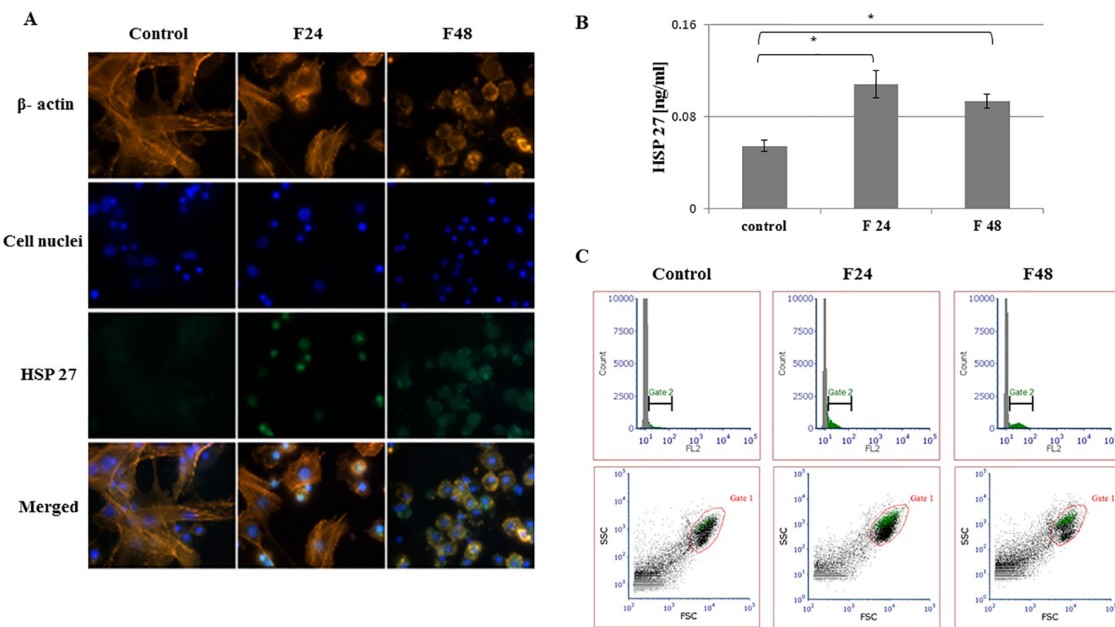

**Fig 4. HSP 27 in hemocytes from *G. mellonella* larvae infected with *C. coronatus*.** (A) Immunofluorescence localization of HSP27 in hemocytes. β- Actin (orange) was stained by ActinRed 555 ReadyProbes Reagent (Invitrogen). Cell nuclei (blue) were stained with Hoechst (Enzo Life Sciences). HSP27 (green) was detected using HSP27 monoclonal antibody (Enzo Life Sciences) and Dylight 488, Goat Anti-Rabbit IgG (Abbkine). (B) HSP27 concentration in hemolymph, as determined by ELISA (Enzo Life Sciences). (C) Flow cytometry data presented as histograms (FITC intensity versus cell count) and dot plots (FSC versus SSC). Gate 1 (red)- all cells; gate 2 (green)- cells containing HSP27. F24- larvae sampled immediately after 24-hour exposure to fungal infection; F48- larvae sampled 24 hours after 24-hour exposure; *p≤0.05; scale bar 25μm.

proteins (HSPs). Under pathological conditions, HSPs begin cytoprotective activities through the initiation of protein folding, repair, refolding of misfolded peptides and possible degradation of irreparable proteins. Increased cellular reactive oxygen species (ROS) levels and subsequent amplified inflammatory reactions result in excessive apoptosis [40]. Since their initial discovery by Ritossa in the salivary glands of *Drosophila melanogaster* larvae [41], HSPs have been detected in animals (vertebrates and invertebrates), bacteria, fungi and plants [42].

Our present findings confirm the presence of four HSPs, *viz.* HSP90, HSP70, HSP60 and HSP27, in the hemocytes of *G. mellonella* larvae. In healthy controls, the highest concentration was recorded for HSP60 and HSP90, whereas HSP70 and HSP27 were found in trace amounts.

A review of the heat shock proteins present in various insect species by Zhao and Jones found the most common forms to be HSP90 and HSP70; however, HSPs were identified in a range of insects from the orders Lepidoptera, Diptera, Hymenoptera, Psocoptera and Hemiptera [43]. One of the few published articles that discuss the presence of HSP in *G. mellonella*, by Wojda et al, indicates that injection of entomopathogenic *Bacillus thuringiensis* cells causes HSP90 levels in larval fat bodies to increase approximately two-fold 1.5 hours after infection, and this rise is followed by a fall to slightly below control levels three hours after infection [44].

Our present findings also confirm that infection influences HSP levels in wax moths, with HSP60 and HSP27 levels remaining elevated in hemocytes 24 and 48 hours after infection with *C. coronatus*. In the F48 group, i.e. 48 hours after infection, an increase of HSP90 was also detected; however, HSP70 levels did not appear to be infected.

Fluorescence microscopy indicates that these proteins are located in adherent cells (plasmatocytes and granulocytes); however, it is difficult to state clearly in which hemocytes subpopulations the HSPs are present, as flow cytometry does not allow the isolation of individual

subpopulations. Such isolation would require the use of antibodies for the specific membrane proteins present in each hemocyte subpopulation, and such antibodies have not been discovered yet. Nonetheless, it can be assumed that in *G. mellonella*, heat shock proteins are involved in the antifungal immune response.

As the synthesis of HSPs increases during infection or tissue damage, they have been a source of interest for immunologists for many years. Most existing studies examine the role of HSPs in mammals. It has been suggested that they represent a link between the innate and adaptive immune systems and that their presence in the circulation serves as danger signals to the host [45]. However, their role in the immune system is more complex. For example, HSPs are involved in antigen presentation, with HSP70 and HSP90 (present in the cytosol) binding antigenic peptides generated within the cell; these antigenic peptides are transported by HSPs to the MHC (major histocompatibility complex) class I molecules present on the cell surface for presentation to lymphocytes [46]. When bound to HSP proteins, pathogen structures are opened to the Toll-like receptors (TLR) located on the surface of cells in the immune system, which can recognize and bind structures characteristic of pathogens. The attachment of a ligand to a TLR stimulates various signal cascades resulting, among others, in the activation of the transcription factor NF-κB.

NF-κB regulates gene expression in response to both stress and infection; however, its activation requires the phosphorylation of the IκB inhibitor by the IKK (IκB kinase) complex, which in turn is stabilized by the heat shock proteins HSP70 and HSP90.

HSPs also participate in signal transduction through the TLR pathway by influencing the stimulation of macrophages. These cells are important components of innate immunity: they produce inflammatory cytokines (e.g. TNFα, IL-1), display phagocytic activity and are involved in stimulation-specific immunity [45, 47–49]. HSPs are also reported to induce the maturation of dendritic cells, as demonstrated by the up-regulation of MHC class I and II molecules. HSP90, HSP70 and HSP60 are also reported to stimulate the proliferation, migration, and cytolytic activity of NK cells, as well as antigen-dependent T cell activation and the production of IFN-γ, and T cell adhesion to fibronectin via TLR2 [50–53]. The effects of HSP27 on human monocytes are predominantly anti-inflammatory, acting through preferential interleukin IL-10 induction and by the alteration of monocytes to immature dendritic cells or to macrophages [54].

Little is known about the role of heat shock proteins in invertebrates. Insects possess an innate type of immunity which shows some structural and functional similarity to that used by mammals; however, although some kind of immunological memory may exist in insects, it is different from that found in mammals.

In insects, the immune response is regulated by three major pathways, *viz*. Toll, Imd and JAK-STAT, which are analogous to certain pathways in the human immune system. The Toll pathway, initially identified in *D. melanogaster*, involves signaling to NF-κB and plays an essential role in embryonic development and immunity [27]. The Toll transmembrane receptor is activated by an extracellular cytokine-like polypeptide that has undergone proteolytic cleavage [55]; as well as by the *Drosophila* Persephone protease following cleavage by fungal virulence factors [56]. The Imd pathway induces the activation of the Dif and Relish transcription factors (NF-κB homologs), which then translocate to the cell nucleus and induce the expression of immune peptides at the genetic level [57, 58].

The JAK-STAT signaling pathway controls inflammation and the activation of leukocytes, such as neutrophils and macrophages [59]. Although it was initially only regarded as a component of human immunity, current studies are increasingly turning to insect models, and those using invertebrates in general, to examine its roles and those of its regulators *in vivo*. Following activation by the ligand upd3, JAK-STAT induces the production of other proteins, including

cytokines and stress response proteins by the fat-body. In turn, upd3(cytokine-like proteins called unpaired 3) is produced by hemocytes in response to various stress conditions, such as injury, heat-shock or dehydration [60].

Some reports suggest that HSPs play a role in immunity in insects. Breeding immunized *G. mellonella* larvae under conditions of moderate thermal shock, or exposure to short-term heat shock before infection, positively affects the immunological activity of hemolymph and expression of immune peptides [61, 62].

## Conclusion

In conclusion, it should be emphasized that heat shock proteins play an important role in the regulation of the immune response in mammals and insects. It can be assumed that the expression of HSP proteins improves the resistance of insects to infection. However, further studies are required to understand the exact mechanism of action of these proteins during fungal infection.

## Supporting information

**S1 Table. Percentage of cells containing the heat shock proteins 90, 70, 60 and 27—Data from flow cytometry.** Gate 1- all cells, gate 2- cells that contain HSP; F24- larvae sampled immediately after 24-hour exposure to fungal infection; F48- larvae sampled 24 hours after 24-hour exposure.
(XLSX)

**S2 Table. Concentration of HSP 90, 70, 60 and 27 in *G. mellonella* hemolymph tested by ELISA—Raw data.** SD- standard deviation; F24- larvae sampled immediately after 24-hour exposure to fungal infection; F48- larvae sampled 24 hours after 24-hour exposure.
(XLSX)

## Author Contributions

**Conceptualization:** Anna Katarzyna Wrońska.

**Data curation:** Anna Katarzyna Wrońska.

**Investigation:** Anna Katarzyna Wrońska.

**Methodology:** Anna Katarzyna Wrońska, Mieczysława Irena Boguś.

**Supervision:** Mieczysława Irena Boguś.

**Visualization:** Anna Katarzyna Wrońska.

**Writing – original draft:** Anna Katarzyna Wrońska.

**Writing – review & editing:** Mieczysława Irena Boguś.

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
