## [Decision Letter · Decision Letter 0]

27 Dec 2019

PONE-D-19-27406

Heat shock proteins (HSP 90, 70, 60, and 27) in Galleria mellonella (Lepidoptera) hemolymph are affected by infection with Conidiobolus coronatus (Entomophthorales).

PLOS ONE

Dear Dr Wrońska,

Thank you for submitting your manuscript to PLOS ONE. After careful consideration, we feel that it has merit but does not fully meet PLOS ONE’s publication criteria as it currently stands. Therefore, we invite you to submit a revised version of the manuscript that addresses the points raised during the review process.

We would appreciate receiving your revised manuscript by Feb 10 2020 11:59PM. To enhance the reproducibility of your results, we recommend that if applicable you deposit your laboratory protocols in protocols.io, where a protocol can be assigned its own identifier (DOI) such that it can be cited independently in the future. For instructions see: http://journals.plos.org/plosone/s/submission-guidelines#loc-laboratory-protocols

We look forward to receiving your revised manuscript.

Kind regards,

Linsheng Song, Ph.D.

Academic Editor

PLOS ONE

Journal Requirements:

We note that one or more of the authors are employed by a commercial company: Biombio.

Reviewers' comments:

Reviewer's Responses to Questions

**Comments to the Author**

1. Is the manuscript technically sound, and do the data support the conclusions?

Reviewer #1: Yes

Reviewer #2: Partly

2. Has the statistical analysis been performed appropriately and rigorously? 

Reviewer #1: Yes

Reviewer #2: Yes

3. Have the authors made all data underlying the findings in their manuscript fully available?

Reviewer #1: Yes

Reviewer #2: Yes

4. Is the manuscript presented in an intelligible fashion and written in standard English?

Reviewer #1: Yes

Reviewer #2: Yes

5. Review Comments to the Author

Reviewer #1: This study was to investigate whether infection by C. coronatus in G. mellonella 24 hemolymph is accompanied by an increase of HSP90, HSP70, HSP60 and HSP27. The results are interesting. And the design is scientific. Some small error should be revised.

1. Line 122: spp should not be italic.

2. Line 134: Please give Sabouraud agar medium a reference.

3. All "ml" should be "mL".

Reviewer #2: In this manuscript Wrońska and co-workers have done a search for the effects of C. coronatus infection on HSPs levels in G. mellonella hemolymph. As a result, the authors identified an increase of HSP60 and HSP27 in hemolymph 24 and 48 hours compared to control group, while HSP90 was increased only in F48. Infection had no observed effect on HSP70 levels. There are methodological points that require clarification and revisions in manuscript.

1 - (Lines 65 and 66): “because it allows the temperatures inside the body of the mammals to be mapped.” The statement regarding mammals has no connection with G. mellonella growth condition mentioned in the previous line.

2 - (Line 133): “control group was formed by larvae exposed for 24 hours”. How study addresses the difference between F24 to F48 groups caused by the stress of temperature change between growth and infection conditions (30 and 20º C respectively)?

3 - (Line 134 / 135): “After exposure, the insects were transferred to new, clean Petri dishes with appropriate food (an artificial diet [37]) and kept for one day”. The temperature kept at this part of the experiment is not clear.

4 - In the methods section (line 132), the number of larvae per Petri dish was informed to be around 15. It is not clear that groups are represented by Petri dishes. The exact number of larvae per group is needed.

5 - In lines 326 and 327 authors state that previous studies show that high larvae mortality rate was 79 and 92% for 24 and 48 hours respectively. In the present study, were included only live larvae or dead were also analyzed? Authors need to inform in methods the criteria to include or exclude a larva.

6 - (Line 227): HSP60 is referred in Fig1 which actually is the HSP90 figure.

7- (Lines 318 and 319): “Insects are often used in biological research and can replace mammals in studies on the immune system or host-pathogen interactions” - The study has no data that supports the statement.

6. PLOS authors have the option to publish the peer review history of their article (what does this mean?). If published, this will include your full peer review and any attached files.

Reviewer #1: No

Reviewer #2: No

---

## [Author Response · Author response to Decision Letter 0]

5 Jan 2020

Dear Editor,

We would like to thank you and the reviewers for taking the time to review our manuscript and for offering such valuable comments. We have addressed these comments in the new version of the text. A summary of the changes is given below.

Reviewer #1: This study was to investigate whether infection by C. coronatus in G. mellonella 24 hemolymph is accompanied by an increase of HSP90, HSP70, HSP60 and HSP27. The results are interesting. And the design is scientific. Some small error should be revised.

1. Line 122: spp should not be italic.

Revised. The font has been changed. 

2. Line 134: Please give Sabouraud agar medium a reference.

Sabouraud agar medium was purchased from Merck. Appropriate information was added to the manuscript. 

3. All "ml" should be "mL".

The abbreviation was revised.

Reviewer #2: In this manuscript Wrońska and co-workers have done a search for the effects of C. coronatus infection on HSPs levels in G. mellonella hemolymph. As a result, the authors identified an increase of HSP60 and HSP27 in hemolymph 24 and 48 hours compared to control group, while HSP90 was increased only in F48. Infection had no observed effect on HSP70 levels. There are methodological points that require clarification and revisions in manuscript.

1 - (Lines 65 and 66): “because it allows the temperatures inside the body of the mammals to be mapped.” The statement regarding mammals has no connection with G. mellonella growth condition mentioned in the previous line.

The sentence: “This feature is useful in immunological research because it allows the temperatures inside the body of the mammals to be mapped” was formulated imprecisely. G. mellonella larvae can be grown in a wide range of temperatures (18 - 37°C). Mapping the conditions in the body of mammals requires a temperature of 37°C. 

The above sentence has been changed to: “This feature is useful in immunological research because it allows to conduct tests at a temperature of 37°C prevailing in the body of mammals.”

2 - (Line 133): “control group was formed by larvae exposed for 24 hours”. How study addresses the difference between F24 to F48 groups caused by the stress of temperature change between growth and infection conditions (30 and 20º C respectively)?

G. mellonella shows normal growth and development in the 18-37°C temperature range. Accordingly, these insects become a model system for assessing host-pathogen interactions. The optimal growth temperature for C. coronatus is 20°C. At other temperatures the virulence against insects of this fungus decreases significantly, which makes conducting experiments impossible. It is likely that enzymes that degrade insect cuticles are produced at this temperature. That is why G. mellonella is the best model in research with this fungus. Both groups F24 and F48 and control were grown at 20°C during the experiment. Therefore, we believe that the effect of temperature on HSP levels did not affect the accuracy of the results. Due to the culture requirements of the fungus, a different temperature could not be used.

3 - (Line 134 / 135): “After exposure, the insects were transferred to new, clean Petri dishes with appropriate food (an artificial diet [37]) and kept for one day”. The temperature kept at this part of the experiment is not clear.

The larvae were kept at 20ºC so that the conditions were unchanging. Appropriate information was added to the manuscript. 

4 - In the methods section (line 132), the number of larvae per Petri dish was informed to be around 15. It is not clear that groups are represented by Petri dishes. The exact number of larvae per group is needed.

I apologize for the inaccuracy, exactly 15 larvae were placed on one Petri dish. The word "around" has been removed from the manuscript

5 - In lines 326 and 327 authors state that previous studies show that high larvae mortality rate was 79 and 92% for 24 and 48 hours respectively. In the present study, were included only live larvae or dead were also analyzed? Authors need to inform in methods the criteria to include or exclude a larva.

Hemolymph was collected from both live and dead larvae. In this experiment, the larvae had a mortality rate of 65±5,6 % and 87±4,8% in F24 and F48 groups respectively. Appropriate information was added to the manuscript.

6 - (Line 227): HSP60 is referred in Fig1 which actually is the HSP90 figure.

Of course, Fig 1 applies to HSP 90. I apologize for the error in the manuscript text, it has been corrected. 

7- (Lines 318 and 319): “Insects are often used in biological research and can replace mammals in studies on the immune system or host-pathogen interactions” - The study has no data that supports the statement.

This sentence does not apply directly to our research. In the introduction and discussion, numerous insect studies are shown, which show that many processes in their bodies are similar to those in mammals. Our research has shown that fungal infection affects heat shock proteins in insects. The same relationship in mammals is described in the literature. Therefore, we wanted to emphasize that in the future, after many more experiments, it may be possible to develop insect models for research on selected mammalian physiological processes.

---

## [Editor Report · Decision Letter 1]

21 Jan 2020

Heat shock proteins (HSP 90, 70, 60, and 27) in Galleria mellonella (Lepidoptera) hemolymph are affected by infection with Conidiobolus coronatus (Entomophthorales).

PONE-D-19-27406R1

Dear Dr. Anna Katarzyna Wrońska,

We are pleased to inform you that your manuscript has been judged scientifically suitable for publication and will be formally accepted for publication once it complies with all outstanding technical requirements.

With kind regards,

Linsheng Song, Ph.D.

Academic Editor

PLOS ONE

---

## [Editor Report · Acceptance letter]

22 Jan 2020

PONE-D-19-27406R1 

Heat shock proteins (HSP 90, 70, 60, and 27) in *Galleria mellonella *(Lepidoptera) hemolymph are affected by infection with *Conidiobolus coronatus* (Entomophthorales). 

Dear Dr. Wrońska:

I am pleased to inform you that your manuscript has been deemed suitable for publication in PLOS ONE. Congratulations! Your manuscript is now with our production department. 

With kind regards,

on behalf of

Dr. Linsheng Song 

Academic Editor

PLOS ONE